# Strength and VO_2_max Changes by Exercise Training According to Maturation State in Children

**DOI:** 10.3390/children9070938

**Published:** 2022-06-22

**Authors:** Liliana Aracely Enríquez-del-Castillo, Andrea Ornelas-López, Lidia G. De León, Natanael Cervantes-Hernández, Estefanía Quintana-Mendias, Luis Alberto Flores

**Affiliations:** Faculty of Physical Culture Sciences, Autonomous University of Chihuahua, Campus II, Periférico de la Juventud y Circuito Universitario S/N. Fracc. Campo Bello, Chihuahua 31125, Mexico; lenriquez@uach.mx (L.A.E.-d.-C.); andreaornelasl4897@gmail.com (A.O.-L.); gdeleon@uach.mx (L.G.D.L.); ncervantes@uach.mx (N.C.-H.); esquintana@uach.mx (E.Q.-M.)

**Keywords:** biological maturity, childhood, child development, aerobic capacity, muscular strength

## Abstract

The health benefits of physical activity (PA) are widely recognized; however, biological maturation contributions are a subject that has been little studied, which is why the aim of this study was to analyze the effect of a six-week training program at moderate-intensity on the muscular strength and aerobic capacity in children between nine and 13 years (13 ± 1.0 years) according to their maturation state. Twenty-six schoolchildren (15 girls) participated in a six-week physical exercise program based on aerobic/anaerobic capacity and coordination skills. Maximal oxygen uptake (VO_2_max), trunk-lift, push-ups, curl-ups, and handgrip strength (both hands) were measured as response variables. Body mass index (BMI), skeletal maturity indicator (SMI), peak height velocity (PHV), age on peak height velocity (APHV) and sex were considered as covariates. The results of VO_2_max, push-ups, curl-ups, and handgrip strength were higher after the exercise program in the whole group (*p* < 0.05). The VO_2_max showed a greater increase in the normal-weight than in the overweight-obesity children (*p* = 0.001). Higher results in dominant handgrip strength were observed in girls (*p* = 0.003). The PHV before intervention presented a positive correlation with the dominant handgrip strength in all kids (r = 0.70, *p* = 0.001). As a conclusion, the six-week training program improved the physical fitness of children independent of the maturation state. Somatic maturation increases the physical abilities in schoolchildren.

## 1. Introduction

The lack of physical activity is one of the main risk factors for the development of cardiovascular, endocrine, and metabolic alterations in children and adolescents. In México, at least, 77% of children spend two hours a day on screen-activities and 82.8% of adolescents between 10 and 14 years of age are considered as inactive since they do not comply with the World Health Organization’s (WHO) recommendations of performing at least 60 min of moderate-vigorous physical activity per day, according to the results of National Health and Nutrition Survey, or ENSANUT, its acronym in Spanish [1]. Likewise, a decrease in the physical activity patterns from childhood to adolescence has been found. A recent meta-analysis of 52 studies with a total of 22,000 participants ranging in ages from three to 18 years, revealed that the greatest decline in moderate-vigorous physical activity occurred at nine years of age, which indicates the importance of promoting healthy lifestyles starting at an early age [2].

Physical activity in children and adolescents is widely recommended by leading organizations such as the WHO, the American College of Sports Medicine (ACSM), and the National Strength and Conditioning Association (NSCA). It has been demonstrated that the implementation of systematized physical activities based on moderate-vigorous aerobic exercises 60-min per session, five days a week, results in physiological adaptations and fitness improvements; additionally, strength and muscular endurance exercises should be incorporated at least three times per week due to their effects on muscle and bone at a young age [3].

However, most of the programs that have shown benefits at those ages are based on high-intensity activities. In a recent systematic review carried out on exercise programs for children and adolescents, it was observed that high-intensity interval training (HIIT) at intensities greater than 90% of the maximum heart rate, performed two to three times per week, with a minimum duration of seven weeks, had more benefits than continuous exercises [4]. In other meta-analysis, a larger effect size was found in an improved aerobic capacity with exercise programs based on HIIT modality than with continuous moderate-intensity exercise [5]. Nevertheless, as benefits are maximized at high intensities, risks could also be potentiated. 

In both reviews, physical activity programs are commonly based on only one type of exercise such as cycling, running, shuttle runs, dance class, or circuit training [4,5]. Thus, it is important to design programs that include different components of physical activity prescription, according to sensitive phases for the development of different physical and motor capacities, with a gradual intensity progression [6].

On the other hand, during early ages, the maturation process is an important factor that is present and could be linked to physical activity benefits as the maturation state shows wide variations in the same chronological age. It is possible to observe 8-year-old children with a skeletal age that can vary four years [7]. At this point, the maturation tempo or growth velocity could represent an important factor in physical performance. Drenowatz et al. [8] observed a maturation state linked to physical fitness after controlling for chronological age. In addition, Fairclough and Ridgers [9] found an influence of maturity status on the physical activity self-perception, where a higher self-perception was found in early mature boys and late mature girls; similar to the outcomes in Cumming et al., where early mature girls presented a lower physical self-worth [10].

However, there is limited evidence to identify how much the maturation tempo can improve the physical fitness of children who are in a training program. Most intervention studies have used sexual maturation as biological maturation, which in fact is a maturation timing indicator, but not a growth velocity indicator or tempo. 

Therefore, the objective of the present study was to analyze the effect of a six-week training program at moderate-intensity on the muscular strength and aerobic capacity in children between nine and 13 years of age according to their maturation state.

## 2. Materials and Methods

### 2.1. Ethical Considerations

The present study was approved by the Ethics Committee of the *Hospital Ángeles de Chihuahua*, Mexico, and was carried out based on the Declaration of Helsinki guidelines and with adherence to the General Law of Health in Research of Mexico [11]. All participants granted their informed consent of voluntary participation and the parents/guardians signed the respective informed written consent.

### 2.2. Study Sample

A sample of 35 children aged nine to 13 years (11 ± 1.0 years) from two public elementary schools in Chihuahua, Mexico, participated in a six-week physical exercise program. However, only 26 children (15 girls) met the inclusion and exclusion criteria. Inclusion criteria were (a) voluntary participation and informed consent; (b) 80% of attendance to the program; and (c) compliance with the assessments before and after the intervention. Exclusion criteria were (a) schoolchildren who were participating in a sport program; (b) who were on a diet program; (c) who had presented a criterion by the Physical Activity Readiness-Questionnaire for Children (PARQ); and (d) the presence of cardiovascular or metabolic diseases as well as muscular, bone, or joint alteration. Previously, the sample size estimate was 13 subjects based on $n=\frac{{\left(Z\;\alpha /2+Z\beta \right)}^{\;2}*S^{2}}{\left(d^{2}\right)}$, where *Z*
*_α_**_/2_* corresponds to the level of statistical significance; *Z*_β_ represents the desired power; *S^2^* represents the variance of VO_2_max changes by a training program; and d represents the difference in the mean (values of VO_2_max pre- and post-intervention, which were taken from a systematic review developed by Flores et al. 2019) [12].

### 2.3. Design

This is a quantitative, quasi-experimental, prospective, and longitudinal study. Upper-body muscular strength and endurance and aerobic capacity were studied as dependent variables; body mass index (BMI), skeletal maturity indicator (SMI), peak height velocity (PHV), age on peak height velocity (APHV), and sex were considered as covariates. All variables were measured before (PRE) and after (POST), a six-week training program was considered as the independent variable.

### 2.4. Physical Fitness Measurements

Muscle strength was measured by static hand dynamometry in both arms, considering the highest record of two attempts, using the A TAKEI^Ⓡ^ model SMEDLEY T-19 digital brand dynamometer. Upper-body muscular endurance was assessed with the Fitnessgram battery’s push-up, curl-up, and trunk lift tests [13]. The push-up and curl-up tests were performed to fatigue guided by a sound cue, while the trunk-lift involved lifting the upper body off the floor with a controlled motion and holding the position. For measurement of the aerobic capacity, the Progressive Aerobic Cardiovascular Endurance Run (PACER) test by Fitnessgram was used, which consists of carrying out the highest number of laps in a 20-m space, guided by a sound indication. The maximal oxygen intake (VO_2_max) was estimated using the equation proposed by Mahar et al. [14], which has been validated for its use in the PACER test in these age groups, quantifying the number of laps they ran.

### 2.5. Body Mass Index

Weight was measured with bioelectric impedance equipment (InBody 230) with 2 to 4 h of previous fasting, which had a sensitivity of 0.1 kg; the height was measured with a stadiometer (Seca 274) with a sensitivity of 0.1 cm placed on the head of the Frankfort plane. Both measurements were made based on the guidelines established by the International Society for the Development of Kineanthropometry [15]; Spanish translation [16] by an ISAK level 2 Anthropometrist, with a technical error of measurement lower than 1%. The BMI was estimated using the weight and stature.

### 2.6. Biological Maturity Estimation

The skeletal age (SA) in years was calculated using the equation by Flores et al. [17], which is based on the percentage of adult stature by the Khamis–Roche method (P-KR) as the predictor variable [18].
Girls SA = (−21.311) + (0.363 ∗ P-KR)
Boys SA = (−17.754) + (0.344 ∗ P-KR)

The skeletal maturity indicator (SMI) was estimated according to the difference between the SA and chronological age, where a difference within ±1.0 year was considered “on-time maturers”, less than −1 year “late maturers”, and greater 1 year “early maturers” [19]. Thus, five children were classified as late, 17 as on time, and four as early maturers.

Maturation offset was predicted according to the sex-specific equation proposed by Mirwald et al. [20], which estimated the temporal distance in years of each participant from their PHV; the difference between chronological age and maturity offset is the predicted APHV. When the difference between chronological age and the APHV was ±0.5 years, it meant that the participant was on their PHV. The maturation offset equation is developed from the leg length, sitting height, stature, weight, and decimal age variables according to the Mirwald method [20]. The sitting height was measured with a Lufkin brand anthropometric tape with a sensitivity of 0.1 cm, and the leg length was established by the difference between the stature and sitting height.

### 2.7. Training Program

The six-week physical exercise program consisted of 60 min-sessions, five days a week, based on the aerobic activities, strength, muscular endurance, and coordination activities. The aerobic capacity was trained in an Exercise Lab 30 min a day, using treadmills, cycle-ergometers, ellipticals, and stair climbers at an intensity based on the heart rate reserve from 40% to 45% for the first three weeks; and from 50% to 60% in the last three weeks. All participants were monitored by a pulsometer (POLAR FT1) during every session. 

Muscle strength and endurance were trained in 30 min-sessions, twice a week for the first three weeks and three times a week for the following three weeks; exercises were based on their body’s own weight and elastic bands; performing up to three to four series of eight to 15 repetitions each. Coordinating activities were worked on during the 30 min-session, three times per week for the first three weeks, then twice a week, using the methodology proposed by Kröger and Roth [21], with activities that promoted basic tactical skills, basic coordination skills, and basic technical skills. At the end of each session, stretching static exercises were performed for five minutes. 

Parents were asked to fill out a 24-h food reminder for four consecutive days: two weekdays and one weekend, before and after intervention, to identify the type, amount, and characteristics of the daily food the children consumed, while the exercise program also served as a measure of variability in consumption during that period.

### 2.8. Statistical Analyses

The Shapiro–Wilk test was used to determine the normality of the data (*p* > 0.05). The means and standard deviations were reported for all variables. The Student’s t-test for independent samples was applied to identify the differences at the baseline measurements by sex, and the Student’s t-test for paired samples was used to identify changes in the physical fitness variables after the training program. The effect size was estimated using the Cohen’s d test where a value greater than 0.5 was considered as a large effect, between 0.2 and 0.5 as a moderate effect and less than 0.2 as a small effect.

A one-way analysis of variance (ANOVA) was performed to identify the association of gender, BMI, SA, SMI, PHV, and physical fitness measurements before the intervention with changes in the physical fitness variables (VO_2_max, handgrip strength, push-up, and, curl-up). A Student’s t-test for independent samples was performed to compare the gain in the VO_2_max as a function of BMI (normal weight vs. overweight/obesity) and the gain in handgrip strength on the dominant arm by sex.

Additionally, ANOVA was used to identify the effect of maturation variables (SA, SMI, PHV, and APHV) on the changes in the physical fitness variables (VO_2_max, handgrip strength, push-up, and curl-up). Finally, a Pearson correlation coefficient between the baseline PHV and handgrip strength gains on the dominant arm was estimated. 

All tests were performed at a 95% confidence level and were developed in SPSS program version 21.0.

## 3. Results

Before the intervention, the girls showed a higher SMI than the boys (0.1 ± 1.0 and −0.6 ± 0.6, respectively, *p* < 0.05); moreover, they were closer to reaching the PHV (−0.8 ± 0.9 and −2.4 ± 1.0, respectively, *p* = 0.001). Regarding the basal values of the physical fitness, only in the trunk-lift test did the girls present a higher value than the boys (24.9 ± 4.5 and 18.4 ± 4.4, respectively, *p* = 0.001) while the boys presented a higher VO_2_max than the girls (47.6 ± 5.5 and 43.5 ± 3.1, respectively, *p* = 0.02). Table 1 shows all of the basal values of each variable by sex.

After the exercise training program, the grip strength increased in the dominant hand to Δ1.46 ± 2.89 kg (15.5 ± 5.1 and 17.0 ± 6.2, respectively, *p* = 0.016), which represented an increase of 9.4%, and the non-dominant hand was Δ1.19 ± 2.53 kg (14.6 ± 4.9 and 15.8 ± 5.3, respectively, *p* = 0.024), which represented an increase of 8.1% in all samples. Furthermore, a significant increase of 42.9% was observed in the curl-up test at Δ3.69 ± 8.38 repetitions (8.6 ± 8.8 and 12.3 ± 9.6, respectively, *p* = 0.034) and the push-up at Δ8.31 ± 5.87 repetitions (7.0 ± 7.6 and 15.3 ± 8.8, *p* = 0.001), which represented greater than 100%; and VO_2_max of Δ3.10 ± 2.27 mL*kg*min (45.2 ± 4.7 and 48.3 ± 5.7, *p* = 0.001), which represented an increase of 6.8% in the whole sample of schoolchildren. Table 2 shows the comparative data before and after the training program.

According to ANOVA, the BMI was associated with the changes in VO_2_max, and sex was associated with gains in the dominant handgrip strength as a result of the physical training program. Table 3 shows the model summaries.

The VO_2_max increased post intervention in normal-weight children to Δ4.79 ± 1.72 mL*kg*min (n= 10, 47.1 ± 3.9 mL/kg/min vs. 51.8 ± 3.9 mL/kg/min, pre/post, respectively, *p* < 0.001) with an effect size of 1.20; and in overweight-obese children Δ1.61 ± 1.91 mL*kg*min (n = 12, 42.5 ± 4.1 mL/kg/min vs. 44.1 ± 4.9 mL/kg/min, pre/post, respectively, *p* = 0.014,) with an effect size of 0.35. However, a greater increase in VO_2_max was observed in the normal-weight group than the overweight-obesity group (*p* = 0.001).

Regarding the changes in the dominant handgrip strength, only the girls showed gains of Δ2.75 ± 0.71 kg (15.8 ± 4.8 kg vs. 18.4 ± 5.6 kg, pre/post, respectively, *p* = 0.003,) with an effect size of 0.5 in comparison to the boys (15.1 ± 5.7 vs. 15.0 ± 6.6 kg, pre/post, respectively, *p* = 0.9).

Finally, in terms of somatic maturation, a direct correlation was observed between the basal PHV and dominant handgrip strength gains with a Pearson coefficient of 0.70 (*p* <0.001) (see Figure 1).

ANOVA was used to examine the effect of SA, SMI, PHV, and APHV on the handgrip strength, push-up, curl-up, and VO_2_max gains, but no significant results were found (>0.05). Additionally, a correlation analysis was performed between the pre- and post-SA differences with respect to the increment in each of the physical fitness variables after the intervention, but no significant relationship was found (*p* > 0.119). As a result, the variations in physical fitness during the intervention period were not attributed to the differences in SA.

## 4. Discussion

In the present study, an increase in aerobic capacity and muscular strength was found in children from nine to 13 years old as a result of a six-week moderate intensity physical activity program with an accumulation of 300 min per week. According to the new physical activity guidelines established by the WHO, the prescription of physical activity in children and adolescents is based on moderate-vigorous activities with gradual increases in the intensity, duration, and frequency, as the present program was designed. 

In a recent systematic review, cardiovascular benefits were observed with high-intensity interval training (HIIT) greater than 90% of the maximum heart rate, two to three times per week, with a minimum duration of seven weeks at those ages [4]. Another recent meta-analysis found that HIIT-based exercise programs had a greater effect size on improving the cardiorespiratory capacity than continuous moderate-intensity exercise programs [5]. However, the results of the current study suggest that the benefits in physical fitness can appear from the first weeks of training, which are generally based on moderate intensities.

Childhood and adolescence represent a critical period for the development of motor skills, so the inclusion of exercises based on coordination capability is important [6,21,22]. In the present research, a short-term exercise program that followed the WHO’s new physical activity guidelines and included coordination activities showed a positive effect on the fitness of children; as a result, it is recommended that any early physical activity program provide an adequate balance of aerobic and anaerobic workouts as well as coordination skills.

Furthermore, the WHO [3] has established that exercise in children and adolescents has offered a positive impact on their musculoskeletal development, fitness, and motor skills, but also provides benefits at the metabolic, cardiovascular, and pulmonary levels. Although the presence of cardio-metabolic diseases associated with lifestyle is uncommon in children, risk markers such as obesity, hypertriglyceridemia, hyperinsulinemia, hyperglycemia, and high blood pressure are present [22], so the proper prescription of physical activity is important.

On the other hand, the present work is one of the first studies to find an increase in physical fitness, regardless of the SMI, as a result of a short-term physical intervention program. Most studies based on physical interventions at a young age do not consider SMI; instead, sexual maturation, which identifies the appearance of secondary sexual characteristics and the onset of puberty, has been commonly used, but no increments in physical fitness have been reported. In a recent systematic review, an increase in VO_2_max/VO_2_peak in prepubertal subjects (determined exclusively by sexual maturation) was observed in eight out of 11 selected studies because of the effect of physical training [12]. In that systematic review, VO_2_max/VO_2_peak increased between 7 and 8% in the analyzed articles, similar to the present study, where the gains of VO_2_max were 7%. In a non-systematic review, Baquet et al. observed an increase in the VO_2_peak between 5 and 6% in prepubescent and pubescent children/adolescents [23]. With respect to the hand-grip strength, a similar study observed an increase of 12.6% in children by a 6-week resistance training program, which was slightly higher than the results of the current study, which were 9.1% for the dominant hand and 8.1% for the non-dominant hand [24].

Most of the investigations that have studied fitness development and the skeletal maturation state have presented a cross-sectional design. Gastin and Bennett [25] observed greater aerobic and anaerobic capacity in early maturers compared to late maturers. Similar results have been reported in rugby football adolescents, where early maturing presented a better ball throw [26]. Pullen et al. (2022) found a relationship of change in maturity with standing long jump performance in girls [27]; Živković et al. (2022) found that a significant interaction of gender and biological maturity was noted for countermovement jump [28]; and in Albaladejo-Saura (2022) as a maturity compensation difference between the genders of adolescent volleyball players related to the physical capacities of power and strength in boys and flexibility in adolescent girls [29]. Likewise, in a meta-analysis carried out by Albaladejo-Saura et al. [30] that included 13 studies, early maturers had higher physical fitness.

Biological maturity status is associated with physical fitness, so training programs should be according to the maturation state [25,26,31]. Additionally, it has been observed in longitudinal studies that late maturers have lower levels of physical fitness before a physical intervention program than early maturers, but showed more increase in fitness over time compared to early maturers [26,32].

However, most of the studies that have examined the relationship between fitness and maturation state, have been carried out for the detection and selection of sport talents. Therefore, it is important to extend this line of research in the health context and not only to the practice of sports, even in schoolchildren with some risk factors such as obesity, due to the importance of the growth, maturation, and physical fitness in terms of health at an early age.

Another key finding of the present study was the direct association between the PHV and the handgrip strength improvement. These results may be partially explained by changes in muscle architecture during the somatic maturation process when PHV is reached [33]. Similar results were published by Nevill et al. [31], where a greater grip strength was found as the PHV was approached. On the other hand, Dobbs et al. [34] observed an increase in the isometric strength of children who achieved their PHV, while those who had not reached the PHV showed no increments when performing a 12-week training program. 

As was explained above, this study is one of the first to consider the somatic and skeletal maturation effects on the fitness variables modified by a training program, however, there were some constraints. One of the most significant limitations of the present research was the lack of a control group to attribute all changes to the intervention. However, pre-post studies were validated and provided reliable evidence [35]. Furthermore, leg strength was not assessed in this study; nevertheless, other works have found lower-body strength improvements as a consequence of a training program at an early age [34,36]. Furthermore, because most of the children in this study were on-time maturers, it would be convenient to examine the effect of physical exercise in a large sample size at different stages of maturation. However, despite these limitations, the present study adds an important and significant contribution to the literature regarding the positive role of moderate-intensity physical training over a short period of time and according to the maturation state.

## 5. Conclusions

In conclusion, it was found that physical fitness improved by a short-term training program based on aerobic and anaerobic exercises and coordination skills with independence of the maturation state. In addition, the proximity to the PHV of schoolchildren with normal weight improved the physical abilities such as handgrip strength and aerobic capacity. 

However, it is necessary to continue with this type of research due to the rapid changes in skeletal maturation that occur during normal childhood and/or adolescence, indicating the presence of a maturational spurt.

## Figures and Tables

**Figure 1 children-09-00938-f001:**
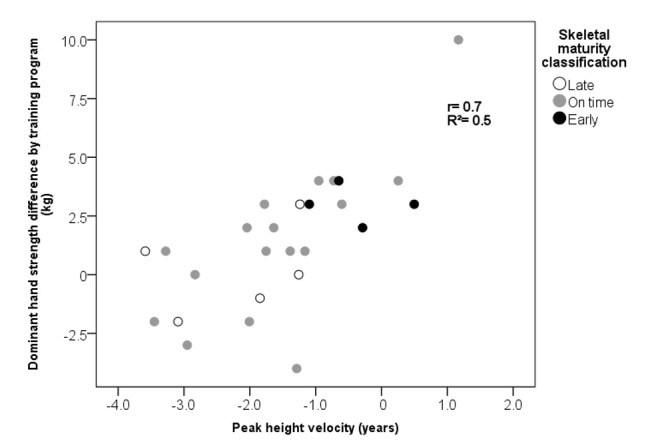
The correlation between the basal PHV and dominant hand strength observed after the training program.

**Table 1 children-09-00938-t001:** The fitness and maturation variables by sex before the exercise program.

Variable	Girls (n = 15)	Boys (n = 11)	*p* Value
	Mean ± SD	Mean ± SD	
Chronological age (years)	11.0 ± 1.0	10.9 ± 0.9	0.838
Skeletal age (years)	11.1 ± 1.5	10.4 ± 1.3	0.148
SMI (years)	0.1 ± 1.0	−0.6 ± 0.6	**0.047**
BMI (kg/m^2^)	19.5 ± 3.0	20.3 ± 5.6	0.919
PHV (years)	−0.8 ± 0.9	−2.4 ± 1.0	**0.001**
APHV (years)	11.9 ± 0.4	13.3 ± 0.3	**0.001**
VO_2_max (mL*kg*min)	43.5 ± 3.1	47.6 ± 5.5	**0.022**
PACER (laps)	17.7 ± 5.5	20.4 ± 9.5	0.489
Trunk-lift (repetitions)	24.9 ± 4.5	18.4 ± 4.4	**0.001**
Push-up (repetitions)	6.0 ± 5.1	8.5 ± 10.1	0.919
Curl-up (repetitions)	10.9 ± 9.4	5.6 ± 7.1	0.281
Hand grip strength D (kg)	15.8 ± 4.8	15.1 ± 5.7	0.646
Hand grip strength ND (kg)	14.8 ± 4.2	14.5 ± 6.1	0.574

SD = standard deviation; SMI = skeletal maturity indicator (difference between the skeletal age and chronological age); BMI = body mass index; PHV = peak height velocity; APHV = age on peak height velocity D = dominant hand; ND = non dominant hand.

**Table 2 children-09-00938-t002:** The fitness and maturation variables before and after the intervention across the whole group (girls and boys).

Variable	PRE	POST	*p* Value	Effect Size
	Mean ± SD	Mean ± SD		
Chronological age (years)	11.0 ± 0.9	11.1 ± 0.9	**0.001**	0.11
SA (years)	10.8 ± 1.4	10.9 ± 1.4	**0.001**	0.07
SMI (years)	−0.2 ± 0.9	−0.2 ± 0.9	**0.008**	0.01
BMI (kg/m^2^)	19.8 ± 4.2	19.8 ± 4.0	0.378	0.01
VO_2_max	45.2 ± 4.7	48.3 ± 5.7	**0.001**	**0.60**
PACER (laps)	18.8 ± 7.4	27.9 ± 11.7	**0.001**	**0.95**
Trunk-lift (repetitions)	22.1 ± 5.5	22.4 ± 6.9	0.797	0.04
Push-up (repetitions)	7.0 ± 7.6	15.3 ± 8.8	**0.001**	**1.02**
Curl-up (repetitions)	8.6 ± 8.8	12.3 ± 9.6	**0.034**	0.40
Hand grip strength D (kg)	15.5 ± 5.1	17.0 ± 6.2	**0.016**	0.26
Hand grip strength ND (kg)	14.6 ± 4.9	15.8 ± 5.3	**0.024**	0.23

SD = standard deviation; SMI = skeletal maturity indicator (difference between skeletal age and chronological age); SA = skeletal age; BMI = body mass index; PHV = peak height velocity; APHV = age on peak height velocity D = dominant hand; ND = non dominant hand.

**Table 3 children-09-00938-t003:** The maximal oxygen intake and dominant handgrip strength explained by sex and BMI. Changes after exercise.

Sources of Variation	Sum of Squares Type III	Fd	F	*p* Value	Partial eta Squared	Observed Power
**VO_2_max (R^2^ = 0.3)**						
BMI basal	27.2	1	6.6	0.017	0.22	0.69
Sex	4.0	1	1.0	0.333	0.04	0.16
Error	95.4	23				
Total	375.8	26				
**Dominant Hand Grip Strength (R^2^ = 0.2)**						
Sex	46.0	1	6.8	0.016	0.22	0.71
Error	162.5	24				
Total	264.0	26				

VO_2_max = maximal oxygen intake; BMI = body mass index. Fd= freedom degree.

## Data Availability

The data are not publicly available to maintain confidentiality. Data are available under justified request to the corresponding author, condition to scientific committee approval.

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
