# Peer review of "Strength and VO2max Changes by Exercise Training According to Maturation State in Children"

_children, 2022, doi:10.3390/children9070938_

Round 1
Reviewer 1 Report
I consider that the sample is too small to reach solid conclusions. The n is not representative of the population.
The aim of the present study was to analyze the effect of a six-week training program at moderate intensity on muscular strength and aerobic capacity in children aged 9 to 13 years according to their maturation stage. In material and methods the authors specify seven weeks of training.
Within the inclusion criteria I consider insufficient the 60% of attendance to the program.
In relation to the material and methods, the FitnessGram PACER test (progressive aerobic cardiovascular endurance running test) has been used to determine the healthy physical condition of students. Currently, the FitnessGram PACER Test Remixes revamps the decades-old fitness test with hip-hop, pop, electronic dance and Latin rhythms, bringing a dance party atmosphere to the gym and motivating students throughout. The new test is a collaboration between The Cooper Institute and Hip Hop Public Health, two organizations dedicated to improving youth fitness, reducing childhood obesity and promoting lifelong healthy habits through research and education.
The ages of the individual children assessed are not specified; only fifteen girls and eleven boys are mentioned. They only specify the mean of both samples.
In relation to the length of the lower limbs, this was established by the difference between the height and the sitting height. From the anthropometric point of view, this measurement is incorrect. The length of the lower limbs is the distance between the trochanteric point and the plane of support, when the subject is standing upright. The sitting height is the distance between the vertex point and the ischial tuberosities, while the subject is sitting on the bench.
We are dealing with an age range of 5 years (9 to 13 years), critical ages for morphological and functional changes. The authors include all schoolchildren in a single group. I think this is due to the small number of schoolchildren evaluated.
No individual data is provided for the schoolchildren, how many were of normal weight or how many were overweight or obese. The authors state that the greatest increase in VO2max was observed in the normal weight group in relation to the overweight-obese group (p=0.001) but they do not specify the n of each subgroup or their ages.
There is no control group. The authors themselves consider that since there is no control group, it is not possible to determine whether the results are entirely due to the intervention.
Of the 31 total bibliographic citations, there are three from 2020 and two from 2021.
Author Response
Subject: Reply letter to reviewers' comments and editorial office's requests
Dear Editorial Board Member of Children Journal,
Please find enclosed the revised version of the manuscript “children-1667656” with the title “Strength and VO2max changes by exercise training according to maturation state in children".
We thank the reviewers for their very helpful suggestions and constructive comments, which have helped us to improve the manuscript. Manuscript was consensually approved.
Our point-by-point responses to all the reviewers’ comments are presented below. Each comment of the reviewer is listed and followed by our responses.
Thanking you in advance for your consideration, we look forward to hearing from you soon.
Yours sincerely,
Luis Alberto Flores Olivares
Correspondence author
Reviewer #1
- Comment 1: “I consider that the sample is too small to reach solid conclusions. The n is not representative of the population”.
Response 1: The sample size is not representative of the population; however, is a sample size number which is commonly used in similar studies.
-Obert P, Courteix D, Lecoq AM, Guenon P. Effect of long-term intense swimming training on the upper body peak oxygen uptake of prepubertal girls. Eur J Appl Physiol Occup Physiol. 1996;73(12):136- 43.
-Nourry C, Deruelle F, Guinhouya C, Baquet G, Fabre C, Bart F, et al. High-intensity intermittent running training improves pulmonary function and alters exercise breathing pattern in children. Eur J Appl Physiol. 2005;94(4):415-23.
-Baquet G, Berthoin S, Dupont G, Blondel N, Fabre C, Van Praagh E. Effects of high intensity intermittent training on peak VO(2) in prepubertal children. Int J Sports Med. 2002;23(6):439 44.
-Lussier L, Buskirk ER. Effects of an endurance training regimen on assessment of work capacity in prepubertal children. Ann N Y Acad Sci. 1977;301:734-47.
-Baquet G, Gamelin FX, Mucci P, Thévenet D, Van Praagh E, Berthoin S. Continuous vs. interval aerobic training in 8-to 11-year-old children. J Strength Cond Res. 2010;24(5):1381-8.
-Gamelin FX, Baquet G, Berthoin S, Thevenet D, Nourry C, Nottin S, Bosquet L. Effect of high intensity intermittent training on heart rate variability in prepubescent children. Eur J Appl Physiol. 2009;105(5):731-8.
-Mandigout S, Lecoq AM, Courteix D, Guenon P, Obert P. Effect of gender in response to an aerobic training programme in prepubertal children. Acta Paediatr. 2001;90(1):9-15.
-Mandigout S, Melin A, Lecoq AM, Courteix D, Obert P. Effect of two aerobic training regimens on the cardiorespiratory response of prepubertal boys and girls. Acta Paediatr. 2002;91(4):403-8.
-Cunha GdosS, Sant’anna MM, Cadore EL, Oliveira NL, Santos CB, Pinto S, et al. Physiological adaptations to resistance training in prepubertal boys. Res Q Exerc Sport. 2014;86(2):172-81.
-George KP, Gates PE, Tolfrey K. The impact of aerobic training upon left ventricular morphology and function in pre-pubescent children. Ergonomics. 2005;48(11-14):1378-89.
-Williams CA, Armstrong N, Powell J. Aerobic responses of prepubertal boys to two modes of training. Br J Sports Med. 2000;34(3):168-73.
- Comment 2: “The aim of the present study was to analyze the effect of a six-week training program at moderate intensity on muscular strength and aerobic capacity in children aged 9 to 13 years according to their maturation stage. In material and methods, the authors specify seven weeks of training”.
Response 2: According to the reviewer's suggestion, the expression “for 7 weeks (page 4)” was removed, it was a mistake.
- Comment 3: “Within the inclusion criteria I consider insufficient the 60% of attendance to the program”.
Response 3: At the beginning of the study design, 60% of program attendance was regarded as an inclusion criterion; nevertheless, the final study sample (26 children) presented more than 83 percent of program attendance. We adjust the inclusion criterion according to the reviewer's suggestion.
- Comment 4: “The ages of the individual children assessed are not specified; only fifteen girls and eleven boys are mentioned. They only specify the mean of both samples.”
Response 4: According to the reviewer's suggestion, age was mentioned in the abstract and methods apartments. It was highlighted in yellow.
- Comment 5: “In relation to the length of the lower limbs, this was established by the difference between the height and the sitting height. From the anthropometric point of view, this measurement is incorrect. The length of the lower limbs is the distance between the trochanteric point and the plane of support, when the subject is standing upright. The sitting height is the distance between the vertex point and the ischial tuberosities, while the subject is sitting on the bench.”
Response 5: Mirwald et al. (2002) validated a method for estimating maturity offset that incorporated the variable leg length, which was calculated as the difference between height and sitting height. We used the Mirwald methodology in this work to estimate maturity offset and age at peak height velocity, therefore we measured/calculated leg length according to the Mirwald procedure.
According to the International Society for the Development of Kinanthropometry (ISAK), the measure to estimate the leg length is the trochanterion height; however, we do not use it, to be attached to the Mirwald methodology.
Mirwald, R. L., Baxter-Jones, A. D., Bailey, D. A., & Beunen, G. P. (2002). An assessment of maturity from anthropometric measurements. Medicine and science in sports and exercise, 34(4), 689-694.
- Comment 6: “We are dealing with an age range of 5 years (9 to 13 years), critical ages for morphological and functional changes. The authors include all schoolchildren in a single group. I think this is due to the small number of schoolchildren evaluated.”
Response 6: Indeed, age 9 to 13 years is a critical period where all children present several differences in morphological, functional, and physical capabilities; however, it was an aspect in favor of calculating the maturity effect on the aerobic capacity and strength changes by the training program.
We are aware that the sample size is a limitation; however, we consider it to have a significant contribution since there are few studies that consider somatic maturation as a control variable when analyzing changes in physical capacities as a result of physical training.
- Comment 7: “No individual data is provided for the schoolchildren, how many were of normal weight or how many were overweight or obese. The authors state that the greatest increase in VO2max was observed in the normal weight group in relation to the overweight-obese group (p=0.001) but they do not specify the n of each subgroup or their ages.”
Response 7: The reviewer's suggestion was attended. We included sample size for normal weight and overweight and obesity children (page 6).
- Comment 8: “There is no control group. The authors themselves consider that since there is no control group, it is not possible to determine whether the results are entirely due to the intervention.”
Response 8: Indeed, one of the most relevant limitations of the present work was the lack of a control group, as we pointed out in the discussion section. At the beginning of the design, we considered a control group; however, the final measurements could not be made due to the covid-19 lockdown, which was the main prevention measure in March 2020 in Mexico. On the other hand, one of the principal factors related to fitness development during the childhood period is the growth and maturity process. For this reason, we used an analysis of variance to examine the maturity effect on the fitness variable changes.
- Comment 9: “Of the 31 total bibliographic citations, there are three from 2020 and two from 2021.”
- Response 9: According to comment nine, which emphasizes that only three citations are from 2020 and 2021, we took on the task of searching for more recent information that discussed the subject of fitness and maturation, for which three more citations from the year 2022 were added, published in the Children Journal, accounting for a total of 18.1% of all citations in the last three years.
All the manuscript changes in this second review were highlighted in yellow.

Reviewer 2 Report
Thank you very much for the opportunity to review this manuscript.
I believe that the topic addressed in the manuscript is of interest; however, the absence of a control group is a very important limitation, which, although it is included in the limitations, substantially affects the validity of the data obtained in the study.
Introduction
In my opinion, the introduction is adequate.
Materials and methods
Ethical considerations
In my opinion, this section is correct.
Study Sample.
It is necessary to include the sample calculation or statistical power with the sample included in the manuscript.
Design
In my opinion, this section is correct; however, the most adequate design should include a control group.
Physical fitness measurements
The minimum real change of this measurement is not included, it is necessary to include it to be able to interpret the data adequately, it can only be considered that there has been a change if it exceeds the minimum real change.
Body Mass Index.
The minimum real change of this measure is not included, it is necessary to include it in order to be able to interpret the data adequately, it can only be considered that there has been a change if it exceeds the minimum real change.
Biological maturity estimation.
The minimum real change of this measure is not included, it is necessary to include it in order to be able to interpret the data adequately, it can only be considered that there has been a change if it exceeds the minimum real change.
Training program
In my opinion, this section is adequate, although all the training could be detailed in detail to make it easier to replicate.
Statistical analyses
In general, this section is correct, however, when performing several tests I propose to perform the Bonferroni correction to reduce the probability of committing a type I error.
Results
I suggest reinterpreting the data once the Bonferroni correction has been applied.
Discussion
In the discussion there is no mention of the statistical power of the study with the sample included, this is of great relevance as we have not performed a sample calculation.
Conclusion
In order to know whether the variables studied have really improved, it is important to know whether they have exceeded the minimum real change and to see the results after applying the Bonferroni correction, without this information it cannot be stated that there is a change, nor that it is statistically significant.
Author Response
Subject: Reply letter to reviewers' comments and editorial office's requests
Dear Editorial Board Member of Children Journal,
Please find enclosed the revised version of the manuscript “children-1667656” with the title “Strength and VO2max changes by exercise training according to maturation state in children".
We thank the reviewers for their very helpful suggestions and constructive comments, which have helped us to improve the manuscript. Manuscript was consensually approved.
Our point-by-point responses to all the reviewers’ comments are presented below. Each comment of the reviewer is listed and followed by our responses.
Thanking you in advance for your consideration, we look forward to hearing from you soon.
Yours sincerely,
Luis Alberto Flores Olivares
Correspondence author
Reviewer 2
“Thank you very much for the opportunity to review this manuscript.
I believe that the topic addressed in the manuscript is of interest; however, the absence of a control group is a very important limitation, which, although it is included in the limitations, substantially affects the validity of the data obtained in the study.”
- Comment 1: “It is necessary to include the sample calculation or statistical power with the sample included in the manuscript.”
Response 1: According to the reviewer's suggestion, sample calculation was estimated (pages 2 and 3).
- Comment 2: “In my opinion, this section is correct; however, the most adequate design should include a control group.”
Response 1: One of the most relevant limitations of the present work was the lack of a control group, as we pointed out in the discussion section.
At the beginning of the design, we considered a control group; however, the final measurements could not be made due to the covid-19 lockdown, which was the main prevention measure in March 2020 in Mexico.
On the other hand, one of the principal factors related to fitness development during the childhood period is the growth and maturity process. For this reason, we used an analysis of variance to examine the maturity effect on the fitness variable changes.
- Comment 3: “Physical fitness measurements.- The minimum real change of this measurement is not included, it is necessary to include it to be able to interpret the data adequately, it can only be considered that there has been a change if it exceeds the minimum real change.”
Response 3: We have performed a statistical analysis based on a 95% level of confidence to assume significant differences, as is usually employed in the literature. However, we include the delta value for all the fitness measures. Additionally, in the discussion section, we compare our observed results with other published results in the literature, founding similar outcomes.
For fitness variables, there is not a minimum real change established, however, several studies have shown increases in VO2max ranging from 5% to 28%, depending on the duration, design, volume of training per week, and other methodologies characteristics of the training programs. Of fact, that is a point to discuss in the majority of systematic reviews and meta-analyses in the physical activity sciences.
- Comment 4: “Body Mass Index.- The minimum real change of this measure is not included, it is necessary to include it in order to be able to interpret the data adequately, it can only be considered that there has been a change if it exceeds the minimum real change.”
Response 4: With respect to the BMI, this variable did not present a statistical difference by the intervention program; nevertheless, BMI was not our study object; only was considered the basal measure to compare normal-weight versus overweight/obese children.
- Comment 5: “Biological maturity estimation.- The minimum real change of this measure is not included, it is necessary to include it in order to be able to interpret the data adequately, it can only be considered that there has been a change if it exceeds the minimum real change.”
Comment 5: In the present study, skeletal age was used as biological maturity indicator; and, as we expected, skeletal age increased by 0.1 years during the training program.
Furthermore, we used skeletal age increases as a control variable to analyze its association to fitness variables changes (VO2max, hand-grip strength, curl-up, push-ups).
If our explication does not respond to the reviewer's suggestion, I would appreciate it if you could explain in more detail what you mean.
- Comment 6: “Training program.- In my opinion, this section is adequate, although all the training could be detailed in detail to make it easier to replicate.”
Response 6: According to the reviewer's suggestion, we could add a file as supplementary material where the program is described in detail, if Children Editorial Office agrees.
- Comment 7: “Statistical analyses.- In general, this section is correct, however, when performing several tests I propose to perform the Bonferroni correction to reduce the probability of committing a type I error.”
Response 7: In our study, we used the t-student test for pairwise comparison, on which the Bonferroni test is based in SPSS software (Statistical software that we employed).
Additionally, the Bonferroni test is commonly used for multiple comparisons (Analysis of variance), which was not the case in this study.
- Comment 8: “Discussion.- In the discussion there is no mention of the statistical power of the study with the sample included, this is of great relevance as we have not performed a sample calculation.”
Response 8: According to the reviewer's suggestion, sample calculation was estimated (pages 2 and 3).
All the manuscript changes in this second review were highlighted in yellow.

Round 2
Reviewer 1 Report
The authors have taken most of the suggestions into account.
Reviewer 2 Report
The authors have made the requested modifications. The manuscript can be accepted in its current state.